# Cannabigerol Modulates Cannabinoid Receptor Type 2 Expression in the Spinal Dorsal Horn and Attenuates Neuropathic Pain Models

**DOI:** 10.3390/ph18101508

**Published:** 2025-10-08

**Authors:** Bismarck Rezende, Gabriel Gripp Fernandes, Vitória Macario de Simas Gonçalves, Gabriela Guedes Nascimento, Kethely Lima Marques, Barbara Conceição Costa Azeredo de Oliveira, Yure Bazilio dos Santos, Maria Eduarda Barros de Andrade, Karine Simões Calumbi, Eduardo Perdigão Maia, Luisa Menezes Trefilio, Fernanda Antunes, Fabrícia Lima Fontes-Dantas, Guilherme Carneiro Montes

**Affiliations:** 1Department of Pharmacology and Psychobiology, Roberto Alcantara Gomes Biology Institute (IBRAG), Rio de Janeiro State University (UERJ), Rio de Janeiro 20551-030, Brazil; rezende.bismarck@posgraduacao.uerj.br (B.R.); goncalves.vitoria@posgraduacao.uerj.br (V.M.d.S.G.); nascimento.gabriela_1@graduacao.uerj.br (G.G.N.); marques.kethely@posgraduacao.uerj.br (K.L.M.); oliveira.barbara_1@graduacao.uerj.br (B.C.C.A.d.O.); santos.yure@graduacao.uerj.br (Y.B.d.S.); eduarda.barros.maria@graduacao.uerj.br (M.E.B.d.A.); simoes.karine@graduacao.uerj.br (K.S.C.); perdigao.eduardo@graduacao.uerj.br (E.P.M.); trefilio.luisa@posgraduacao.uerj.br (L.M.T.); 2NeuroPharmacoGenetics Laboratory, Department of Pharmacology and Psychobiology, Rio de Janeiro State University (UERJ), Rio de Janeiro 20551-030, Brazil; gabriel.gripp@ufjf.br; 3Department of Biophysics and Physiology, Federal University of Juiz de Fora, Juiz de Fora 36036-900, Brazil; 4Animal Clinic and Surgery Laboratory, Center for Agricultural Sciences and Technologies (CCTA), State University of Northern Fluminense Darcy Ribeiro (UENF), Campos dos Goytacazes 28013-602, Brazil; fernandaanest@uenf.br

**Keywords:** *cannabis*, pain, cannabigerol, chronic pain, cannabinoid receptor type 2 (CB2R)

## Abstract

**Background/Objectives:** The expanding focus on novel therapeutic pathways for long-term pain relief has directed interest toward compounds obtained from Cannabis sativa. This study evaluated the antinociceptive potential of cannabigerol-enriched extract (CBG) in models of acute and chronic hypernociception, along with morphological outcomes. **Methods:** Formalin and hot plate tests were used on male Swiss mice to assess acute oral antinociception. To the chronic pain model, 8-week-old male Wistar rats underwent spinal nerve ligation (SNL), and CBG was administered orally by gavage once daily for 14 days. **Results:** CBG reduced nociceptive responses in the formalin test and hot plate tests, mainly at a dose of 30 mg/kg, showing antinociceptive activity. CBG attenuated SNL-induced thermal and mechanical hypersensitivity, accompanied by reduced microglial density and spinal morphological changes. Importantly, cannabinoid receptor type 2 (CB2R) signaling contributed to the antinociceptive effects of orally administered CBG, whereas cannabinoid receptor type 1 (CB1R), Brain-Derived Neurotrophic Factor (BDNF), and Tumor Necrosis Factor (TNF) did not appear to play major roles under our experimental conditions. **Conclusions:** Collectively, these findings support CBG as a promising alternative for chronic pain management.

## 1. Introduction

Chronic pain is a leading global health problem, with meta-analyses estimating a prevalence of approximately 20–30% among adults, higher in women and older individuals [1,2]. Systematic reviews also highlight its profound impact on disability, quality of life, and work capacity, making it a major contributor to years lived with disability worldwide [3].

A 2023 systematic review estimated that 35.7% of Brazilian adults suffer from chronic pain, rising among older adults [4]. Neuropathic pain affects around 10% of adults and is associated with greater interference in daily activities compared with non-neuropathic chronic pain [5].

Despite significant advances in understanding pain mechanisms, current therapeutic approaches remain insufficient, particularly for chronic and neuropathic pain. Conventional treatments such as nonsteroidal anti-inflammatory drugs (NSAIDs), antidepressants, gabapentinoids, and opioids often fail to provide adequate long-term relief and are associated with undesirable side effects, including gastrointestinal toxicity, tolerance, and dependence [6]. Among available treatments, opioids carry a pronounced risk of dependence and abuse, which has become a pressing global health problem [6,7].

In this context, cannabinoids have attracted growing interest due to their neuroprotective, antioxidant, anti-inflammatory, and analgesic properties demonstrated in both preclinical and clinical studies [8,9,10,11,12]. A systematic review of pharmacological and non-pharmacological treatments for neuropathic pain, along with French clinical recommendations, highlights the therapeutic relevance of cannabinoid-based approaches as part of comprehensive pain management strategies [13]. Recent clinical trials have reinforced these findings: a phase III, 12-week study demonstrated that a cannabis-based product containing tetrahydrocannabinol, cannabidiol, and cannabinol (THC/CBD/CBN) significantly reduced pain in patients with diabetic neuropathy, with a favorable safety profile [14]. Furthermore, meta-analyses and double-blind, randomized trials indicate benefits and manageable adverse effects of THC/CBD extracts in neuropathic pain [15,16]. Our group has been investigating the endocannabinoid system and recently demonstrated that an oral Cannabis sativa extract rich in cannabigerol (CBG) at 50 mg/kg produced significant antinociceptive effects without impairing locomotor performance in animals exposed to prenatal hypoxia-ischemia [17]. In this context, the present work investigates the analgesic potential of CBG-rich extracts from *Cannabis sativa*, using a reduced oral dose of 30 mg/kg to assess its antinociceptive efficacy in neuropathic pain models and to explore the molecular mechanisms underlying its activity.

## 2. Results

### 2.1. Oral Cannabigerol Produces Central and Peripheral Antinociceptive Effects in Rodent Pain Models

Oral administration of CBG at 30 mg/kg produced significant antinociceptive effects in different experimental pain models. In the formalin test, CBG (30 mg/kg) markedly reduced nociceptive behavior in both the neurogenic and inflammatory phases. Figure 1A illustrates the responses observed during the neurogenic and inflammatory phases, following oral administration of vehicle, morphine, ASA, and CBG in the animals. It was observed that the formalin response time during the neurogenic phase in the CBG-treated group was reduced to 41.5 ± 2.7 s compared with 68.6 ± 6.8 s in the vehicle group. During the inflammatory phase, animals treated with CBG exhibited a decrease in formalin response time from 167.5 ± 32.7 s (vehicle) to 45.5 ± 10.7 s. In the hot plate test, CBG (30 mg/kg) produced %MPE values of 30.8 ± 9.2%, 28.4 ± 9.0%, and 28.2 ± 6.7% at 50, 60, and 70 min, respectively, all significantly higher than the vehicle. In the SNL model, thermal and mechanical hypernociception developed 7 days after surgery (Figure 1C,D), while no hypernociception occurred in the sham group. In male rats, paw withdrawal latency (PWL) decreased from 6.1 ± 0.4 s and 201.1 ± 1.9 g (baseline on day 0) to 2.1 ± 0.1 s and 89.3 ± 5.8 g (day 7) in the thermal and mechanical assessments, respectively. In the SNL + vehicle group, PWL values decreased from 5.8 ± 0.3 s and 198.4 ± 3 g to 2.2 ± 0.2 s and 84 ± 2.4 g, confirming the establishment of thermal and mechanical hypernociception. Treatment with CBG significantly increased latency (*p* < 0.05) in both thermal and mechanical hypernociception assessments when compared to the vehicle-treated group. Reversal of thermal and mechanical hypernociception was observed from the 10th to the 21st day post-surgery.

### 2.2. Cannabigerol Attenuates Microgliosis but Fails to Modulate Astrogliosis

As depicted in Figure 2A–C, CBG was able to block the microgliosis observed in Rexed layers I and II of SNL submitted animals (Figure 2B) associated with a reduction in their soma area, a parameter commonly used to assess microglia reactivity and pro-inflammatory phenotype Figure 2C [18,19,20]. However, CBG did not reverse the elevated TNFα levels detected in the dorsal horn after SNL surgery (E). Additionally, SNL animals displayed marked astrogliosis compared to the sham group, and CBG treatment failed to attenuate this response (Figure 2D). These findings indicate that, in the SNL model, CBG can modulate microglial activation but has limited effects on TNFα upregulation in the spinal dorsal horn.

Among glia cells responsible for the development of neuropathic pain, several reports support the hypothesis that astroglia cells may act in synergy with microglia cells, secreting neurotrophic and pro-inflammatory mediators, ultimately leading to allodynia and hyperalgesia [21]. In our protocol, the SNL submitted rats present astrogliosis compared to the SHAM group, and CBG failed to block this phenomenon (Figure 3), suggesting that microglia cells may be one of the major glia cells responsible for CBG actions.

### 2.3. Involvement of Cannabinoid 2 Receptors in CBG Effects

In order to investigate the molecular mechanism responsible for CBG action, we initially investigated the role played by brain-derived neurotrophic factors (BDNF) in our model, since some reports support the role of this neurotrophin in the genesis of neuropathic pain [22]. BDNF levels remained unchanged in our animal model (Appendix A). Next, we assessed the role of CB1 and CB2 in the mechanism of action of CBG (Figure 4). Using the formalin test, we observed that SR144528, a selective CB2R antagonist, but not AM281, a selective CB1R antagonist, blocked the CBG effect in the neurogenic phase of the formalin test (A) but had no effect in the inflammatory phase (B). To further explore this, we measured the optical density of CB1 and CB2 receptors in laminae I/II of the dorsal horn in the spinal cord of SNL animals treated with CBG (C–F). Interestingly, while we did not observe any changes in CB1 receptor expression, SNL reduced CB2R optical density in the spinal cord, and CBG reversed this effect.

## 3. Discussion

Chronic pain is broadly acknowledged as a critical global health issue, ranking among the foremost causes of disability and generating significant social and economic costs [1,2,3].

In this context, our study demonstrates that oral administration of CBG at 30 mg/kg produces robust antinociceptive effects in both acute and neuropathic pain models. These results reinforce previous findings from our group showing that CBG at 50 mg/kg reduced hyperalgesia induced by prenatal hypoxia–ischemia, without impairing locomotor activity [17]. In that previous study, a higher dose of the same extract was administered, and no signs of sedation or motor impairment were observed in the animals. Pharmacokinetic and safety studies indicate that CBG is well absorbed following oral administration, distributes to the brain, and is tolerated without significant adverse effects [23,24]. These findings support the selection of the 30 mg/kg dose used in the present study and corroborate our observation that CBG does not induce sedation or motor impairment in the tested animals. Therefore, confounding effects related to sedation or motor impairment are unlikely at the dose used in the present study. While the earlier work also reported reductions in TNFα and Nav1.7 expression in the spinal cord, the current data highlight additional effects of CBG in different pain paradigms, including strong inhibition of nociceptive behavior in the formalin test, reversal of thermal and mechanical hypernociception after SNL, and attenuation of microgliosis in the spinal dorsal horn.

Consistent with our observations, preclinical studies in other neuropathic pain models—including chemotherapy-induced peripheral neuropathy—show that chronic administration of CBG maintains efficacy without tolerance. Importantly, combinations of CBG with CBD have been reported to reduce mechanical hypersensitivity more effectively than CBG alone [25,26]. Moreover, systematic reviews [27] corroborate that CBG, whether administered alone, combined with CBD, or as part of CBG-rich extracts, consistently exhibits antinociceptive effects across neuropathic pain models. Collectively, these findings highlight the central and peripheral analgesic potential of CBG, supporting its development as a novel therapeutic agent.

Several lines of evidence suggest an interaction between the endocannabinoid system and the secretion of neurotrophic factors, such as BDNF. In this direction, we investigated whether CBG was able to interact with BDNF. Surprisingly, in our SNL neuropathic pain model, BDNF optical density was unchanged among all groups analyzed. Supporting our observation, Geng and colleagues observed that BDNF decreases to Sham-control levels 14 days after SNL and sustains until 28 days [28]. This effect may account for the lack of differences described in the present study. Nonetheless, the role of CBG in modulating BDNF at other points was not evaluated in the present study, necessitating further experiments to better delineate the interaction between CBG and neurotrophins and their role in generating neuropathic pain.

Beyond our observations, the endocannabinoid system has been described to interact with BDNF in order to promote synaptic plasticity [29]. Specifically, it has been shown that in the striatum, long-term depression (LTD) induced by BDNF was blocked after CB1 antagonist administration [30] and BDNF regulates CB1R expression in this region [31]. In the same direction, CB1R blockage disrupts the cortical interneurons’ LTD facilitated by BDNF administration in a subthreshold protocol [32]. Furthermore, synaptic plasticity has been implicated in the development and maintenance of neuropathic pain [33,34], and several studies outline that neurons present in the dorsal horn are capable of both long-term potentiation (LTP) and LTD [35,36,37]. In this sense, our data suggests that it seems unlikely that CBG is acting through CB1R and BDNF systems to facilitate LTD, ultimately leading to an anti-nociceptive phenotype.

Besides CB1R, pharmacological studies suggest that CBG may interact with CB2R in a higher-affinity manner, acting as an agonist, while acting as a partial agonist with CB1R [38,39,40,41]. In accordance with Sepulveda and colleagues [26], we also demonstrated that CBG reduces nociception through CB2 receptors. It is well established that SNL augments the CB2 levels in the spinal cord at least seven days after surgery [42], which may be related to the fact that the endocannabinoid system is responsible for breaking the inflammation process. Nonetheless, in our study, SNL reduced CB2 receptor levels, consistent with the hyperalgesic phenotype observed in these animals. Indeed, in vitro studies using BV2 cells showed that CB2 receptor expression may increase or decrease according to the inflammatory challenge used or the time point analyzed [43,44], corroborating the discrepancy on CB2 levels in the present study and those observed by Zhang and colleagues [42]. Additionally, during inflammatory events, the endogenous CB2 receptor ligands, such as anandamide (AEA) and 2-arachidonoylglycerol (2-AG), are degraded through the COX-2 alternative pathway, enzymes usually upregulated during inflammatory events [45]. This prompts us to suggest that CBG interacts with CB2, preserving its signaling pathway, leading to an antinociceptive phenotype associated with reduced microgliosis.

To explore the mechanism behind CBG’s effect on microgliosis, we investigated whether it involves a reduction in TNF levels, since several pieces of evidence suggest that TNF facilitates nociception through neuronal plasticity and LTP [46,47]. Unexpectedly, we did not observe any significant reduction in TNF levels in the spinal cord dorsal horn of SNL. The divergent result observed may indicate that CBG acts through an unknown molecular mechanism in our animal model. In fact, it has been shown that LPS-stimulated BV2 produced both TNF and NO, and that CBG blockades the secretion of both mediators [48]. Compelling evidence has also shown that NO plays a crucial role in pain [49]. Borrelli et al. (2013) [50] demonstrated that CBG attenuates murine colitis by reducing nitric oxide production and inducible nitric oxide synthase (iNOS) protein expression in macrophages, an effect modulated by CB2R, along with antioxidant and anti-inflammatory actions. In this sense, further studies are needed to investigate whether CBG has its antinociceptive properties through reduction in NO synthesis via iNOS, an isoform upregulated in inflammatory events [51], or through another, yet undescribed, mechanism. In line with previously demonstrated by our group [52], SNL induced an astrogliosis in the dorsal horn, a phenomena that was not revered by CBG. This fact has been already demonstrated in other animal models. For instance, it has been described that either that treatment with omega-3-enriched fish oil or mitochondrial therapy, produced an antinociceptive effect and anti-inflammatory effect associated with none effect on astrogliosis, suggesting that the anti-inflammatory effect of some drugs may be devoid of an astrocytic-dependent mechanism [53,54]. Strengthening the assumption that astrocytes may not have a sole role in neuropathy in SNL, some evidence demonstrated that central nervous system astrocytes, but not microglia, seems to be the source of TNF-α expression [55], this evidence is in accordance with the present report, where CBG had no effect on either TNF or GFAP stain. It is also important to consider sex-dependent factors that may influence pain responses and cannabinoid mechanisms. In a previous study, CBG reduced TNFα expression in males and Nav1.7 expression in females in a model of prenatal hypoxia–ischemia [17], indicating sex-specific mechanisms in nociception. Although the present study focused exclusively on male animals, future investigations should determine whether similar CB2-dependent effects occur in females to enhance the translational relevance of CBG.

## 4. Materials and Methods

### 4.1. Laboratory Animals

The experimental procedures applied in this research were approved by the Ethics Committee on the Care and Use of Laboratory Animals of the State University of Rio de Janeiro (CEUA/UERJ, protocol 018/2023). The study involved male Swiss mice (*Mus musculus*), aged 4–6 weeks and weighing 25–30 g, as well as male Wistar rats (*Rattus norvegicus*), weighing 180–280 g. All animals were bred and maintained in the animal facility of the Department of Pharmacology and Psychobiology (DFP), Instituto de Biologia Roberto Alcantara Gomes (IBRAG), UERJ. They were kept under standard laboratory conditions: a 12 h light/dark cycle (lights off from 6:00 p.m. to 6:00 a.m.), temperature of 21 ± 1 °C, and relative humidity of 50 ± 2%. Housing was provided in polypropylene cages with sawdust bedding, with groups of four animals randomly assigned. Food in pellet form and drinking water were available ad libitum. Before the beginning of the experiments, the animals were allowed a 30 min acclimatization period at room temperature.

### 4.2. Drugs

The cannabis extracts used in this study were generously provided by the Association for the Support of Research and Medicinal Cannabis Patients (APEPI, Rio de Janeiro, Brazil). The cannabis extract used in this study was characterized at the Center for Information and Toxicological Assistance (CIATox, University of Campinas, Campinas, Brazil) which provided an analytical certificate of composition. According to the report, the extract was highly enriched in cannabigerol (CBG, 59.0% m/m; 95%), while Δ9-THC was present only at 0.5% m/m (0.9%). Cannabidiolic acid (CBDA), cannabidiol (CBD), tetrahydrocannabinolic acid (THCA) and cannabinol (CBN) were below the detection limit (<0.2%). The study also employed acetylsalicylic acid (ASA, Sigma-Aldrich, St. Louis, MO, USA), SR144528, a selective CB2 receptor antagonist, AM281, a selective CB1 receptor antagonist (Sigma, Saint Louis, MO, USA), dimethyl sulfoxide (DMSO, ≥99.9% purity, Dinâmica, Indaiatuba, Brazil), formaldehyde P.A. 36% (Proquimios, Rio de Janeiro, Brazil); xylazine hydrochloride 2% (Vetecia, Louveira, Brazil), morphine and ketamine hydrochloride (Cristália, Itapira, Brazil), and isoflurane (BioChimico, Itatiaia, Brazil). For experimental procedures, the CBG-rich cannabis extract (comprising 95% of the total plant cannabinoids), morphine, ASA, SR144528, and AM281 were freshly dissolved in DMSO immediately before administration. The vehicle consisted of dimethyl sulfoxide (DMSO, ≥99.9% purity), administered orally without dilution (100% *v*/*v*).

### 4.3. Experimental Model

All animals were randomly allocated to treatment groups using a random number generator. Drug solutions were prepared and coded by an experimenter not involved in behavioral assessments. Oral administration and surgical procedures were performed by one investigator, whereas nociceptive scoring and data collection were carried out by a second investigator blinded to group allocation. Treatment codes were revealed only after completion of data analysis to ensure allocation concealment and minimize bias.

#### 4.3.1. Formalin Test

The formalin test was used as a model of acute peripheral nociception [56]. Mice received a subcutaneous injection of 20 µL of 2.5% formalin into the plantar surface of the right hind paw. Nociceptive behavior was assessed in two distinct phases: phase 1 (0–5 min post-injection), representing neurogenic pain, and phase 2 (15–45 min post-injection), representing inflammatory pain. Animals (*n* = 10 per group) were treated orally (p.o.) by gavage with vehicle, morphine (30 mg/kg), CBG (1–30 mg/kg), or ASA (300 mg/kg). Fifty minutes after oral treatment, the formalin injection was administered. Nociceptive responses were quantified as the total time (in seconds) spent licking, scratching, or biting the injected paw during the 45 min observation period. To elucidate the involvement of cannabinoid receptors in the antinociceptive activity of CBG, the antagonists AM281 (1 mg/kg) and SR144528 (10 mg/kg) were administered intraperitoneally (i.p.); subsequently, the animals (*n* = 7) received oral CBG 30 min later, and formalin was injected 30 min thereafter.

#### 4.3.2. Hot Plate Test

To evaluate the analgesic activity (MPE), we employed the hot plate test [57]. Initially, animals were habituated to the hot plate apparatus (51 ± 1 °C). Reaction latency (paw licking, paw shaking, or jumping) was then assessed twice, with a 30 min interval between trials, to establish the control latency. The mice with a reaction time longer than 20 s were excluded. After obtaining control latency, animals (*n* = 10) received oral administration (by gavage) of either vehicle, morphine (30 mg/kg), or CBG (30 mg/kg). Reaction time was recorded from 0 up to 150 min after compound administration, with a cut-off time of 30 s to prevent paw injury. Antinociceptive efficacy was expressed as the percentage of the maximum possible effect (%MPE), calculated according to Equation (1): % MPE (post-drug latency) − (control latency)/(cut-off time) − (control latency) × 100

#### 4.3.3. Chronic Nociception Model

Chronic nociception was induced via spinal nerve ligation (SNL), in accordance with previously established protocols [58]. Briefly, male Wistar rats were anesthetized with an intraperitoneal injection of xylazine (20 mg/kg) and ketamine (100 mg/kg). Following disinfection of the surgical site with 10% povidone–iodine, a midline dorsal incision was made to expose the lateral laminae of the lower lumbar and upper sacral vertebrae. The right L6 transverse process was carefully removed to allow for exposure and tight ligation of the right L5 and L6 spinal nerves using 6-0 silk sutures. Animals in the sham group underwent the same surgical procedure, excluding nerve ligation. Seven days post-surgery, SNL animals exhibited thermal and mechanical hypernociception. At this point, pharmacological treatment was initiated and continued daily for a duration of 14 days.

#### 4.3.4. Evaluation of Thermal and Mechanical Hypernociception in the SNL Model

Seven days after SNL surgery, animals developed both thermal and mechanical hypernociception. They were then allocated into three experimental groups (*n* = 7–8 per group): (1) SNL + CBG (30 mg/kg, orally by gavage), (2) SNL + vehicle (orally), and (3) sham surgery + vehicle (orally). Thermal hypernociception was assessed using the paw immersion test, in which the ipsilateral hind paw was immersed in water maintained at 46 °C. The primary outcome was the latency (s) to paw withdrawal, defined as the thermal nociceptive threshold. A cut-off time of 15 s was set to prevent tissue damage [59]. Mechanical hypernociception was evaluated using a digital analgesiometer (Insight, Brazil), which applies a gradually increasing force to the dorsal surface of the hind paw at a constant rate (grams per second). The applied force was continuously recorded and displayed on an electronic panel. Paw withdrawal in response to the stimulus was considered a nociceptive response, and the corresponding force value (in grams) at the moment of withdrawal was recorded as the mechanical nociceptive threshold [60].

### 4.4. Tissue Collection and Immunofluorescence

A cohort of rats was anesthetized with xylazine (20 mg/kg, i.p.) and ketamine (100 mg/kg i.p.) and transcardially perfused with PBS 0.01M (pH 7.4) followed by PFA 4%. Immediately after, the spinal cord was dissected and transferred to PFA 4% overnight. After, the spinal cords were stored in a 30% sucrose solution for cryoprotection. A week later, brains were frozen in −30 °C isopentane and stored at −80 °C until being processed. Coronal slices, 30 μm thick, were obtained from the lumbar part of the spinal cord using a cryostat at −20 °C (Leica Microsystems, Wetzlar, Germany). The sections were kept in an anti-freezing solution (30% ethylene glycol/20% glycerol) until the immunofluorescence assays were performed.

Briefly, free-floating sections were washed three times with PBS (15 min each) followed by 30 min incubation with citrate buffer 10 mM (pH = 6.0) at 70 °C for antigen retrieval. After three washes with PBS (15 min/wash), the sections were incubated with a blocking solution (TBS + Triton × 0.025% + BSA 3%) for 2 h followed by overnight incubation with anti-Iba1 antibody (rabbit, 1:750; FUJIFILM Wako Pure Chemical Corporation, Osaka, Japan), anti-TNF (rabbit, 1:100; Thermo Fisher Scientific, Waltham, MA, USA), anti-GFAP (mouse, 1:1000; Abcam, Cambridge, UK), anti-BDNF (rabbit, 1:250; Thermo Fisher Scientific, Waltham, MA, USA), anti-CB1R (rabbit, 1:250; Thermo Fisher Scientific, Waltham, MA, USA), and anti-CB2R (rabbit, 1:500; Thermo Fisher Scientific, Waltham, MA, USA). After three washes with PBS, the sections were incubated for 1.5 h with a secondary antibody (anti-rabbit AlexaFluor 488, anti-mouse 555; 1:750). DAPI staining (1:10,000 from 2 μg/mL stock solution) was used to visualize the dorsal horn of the spinal cord and oriented our analysis towards Rexed layers I and II.

### 4.5. Image Acquisition and Quantification

Photographs were taken epifluorescence microscope, Olympus BX53 (Olympus Corporation, Hachioji, Tokyo, Japan). The IBA-1 cell density was manually counted under a 20× objective and calculated by normalizing the number of cells counted by the area analyzed (measured using a 10× objective). Optical density of TNF, GFAP, BDNF, CB1R, and CB2R was calculated using the software FIJI (version 1.53t), after image acquisition under a 10× objective. The GFAP-stained area was estimated using the FIJI automatic plugin in images acquired using a 20× objective.

### 4.6. Statistical Analysis

All statistical analyses and graphical representations were performed using GraphPad Prism^®^ version 10 (GraphPad Software Inc., San Diego, CA, USA). Data normality was evaluated with the Shapiro–Wilk test. For datasets showing a parametric distribution, one-way ANOVA followed by Dunnett’s post hoc test was applied for the formalin test, while two-way ANOVA followed by Sidak’s post hoc test was used for the hot-plate test and assessments of thermal and mechanical hypernociception. Experimental groups were compared to their respective controls. Results are expressed as mean ± standard error of the mean (SEM), and differences were considered statistically significant at *p* < 0.05. Significance levels are indicated by symbols (* or #).

## 5. Limitations of the Study

Although this study provides important insights into the antinociceptive effects of CBG in models of acute and chronic pain, it presents some limitations. As it was conducted in an animal model, the findings may not be directly extrapolated to humans due to physiological and pharmacokinetic differences. Despite evaluating multiple molecular markers (BDNF, CB1R, CB2R, IBA1, TNF, and GFAP) in the spinal cord, analyses were not performed in other relevant tissues, such as the dorsal root ganglion (DRG), which plays a key role in pain modulation. The experimental design employed a specific route, dose, and treatment duration, which may not reflect other clinically relevant regimens. While CB1 and CB2 antagonists were used in a screening test, the involvement of this system and other cannabinoid or non-cannabinoid targets cannot be excluded due to experimental and ethical limitations. Immunohistochemistry provided data on protein expression but did not include functional assays to confirm receptor activity. In addition, the evaluation at fixed time points may not capture the full dynamics of molecular changes during the transition from acute to chronic pain. Another limitation is that only male animals were included, which precludes conclusions on potential sex-dependent differences. Considering that pain mechanisms and cannabinoid responses may differ between sexes, further studies including female subjects are necessary to strengthen the translational relevance of our findings. Therefore, the translational applicability of the results requires further validation. Future studies should investigate the long-term efficacy and safety of CBG in chronic pain models to expand the understanding of its therapeutic potential.

## 6. Conclusions

In summary, our study provides robust evidence that CBG exerts potent antinociceptive effects across acute, inflammatory, and neuropathic pain models. Importantly, we demonstrate that a lower oral dose (30 mg/kg) of a CBG-rich extract is sufficient to produce these effects, enhancing translational relevance compared with previous reports at higher doses. Mechanistically, our findings indicate that CB2 receptor signaling mediates the antinociceptive action of CBG, as supported by receptor restoration after spinal nerve ligation and selective blockade by SR144528, whereas CB1R, BDNF, and TNF did not appear to play a major role under our experimental conditions. Additionally, we show that CBG attenuates microgliosis but not astrogliosis in the spinal dorsal horn, providing new cellular insights. Collectively, these results highlight CBG as a promising candidate for pain management and support further translational studies.

## Figures and Tables

**Figure 1 pharmaceuticals-18-01508-f001:**
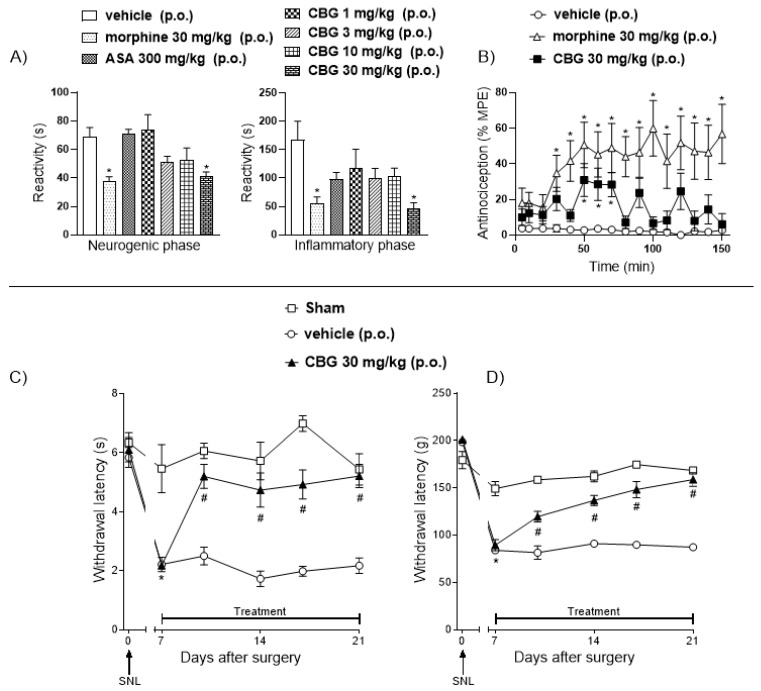
Effects of oral gavage administration of vehicle, morphine (30 mg/kg), ASA (300 mg/kg), or high-CBG cannabis extract (30 mg/kg) on the neurogenic and inflammatory phases of the formalin test (**A**); and on hot-plate test performance (**B**) in male Swiss mice. The effects of a 14-day oral treatment with vehicle or CBG on thermal (**C**) and mechanical (**D**) hypernociception in male Wistar rats, 7 days after SNL surgery. Data are expressed as mean ± SEM. *p* < 0.05. * *p* < 0.05 vs. 0 day (before SNL); # *p* < 0.05 when compared to the vehicle group throughout the treatment.

**Figure 2 pharmaceuticals-18-01508-f002:**
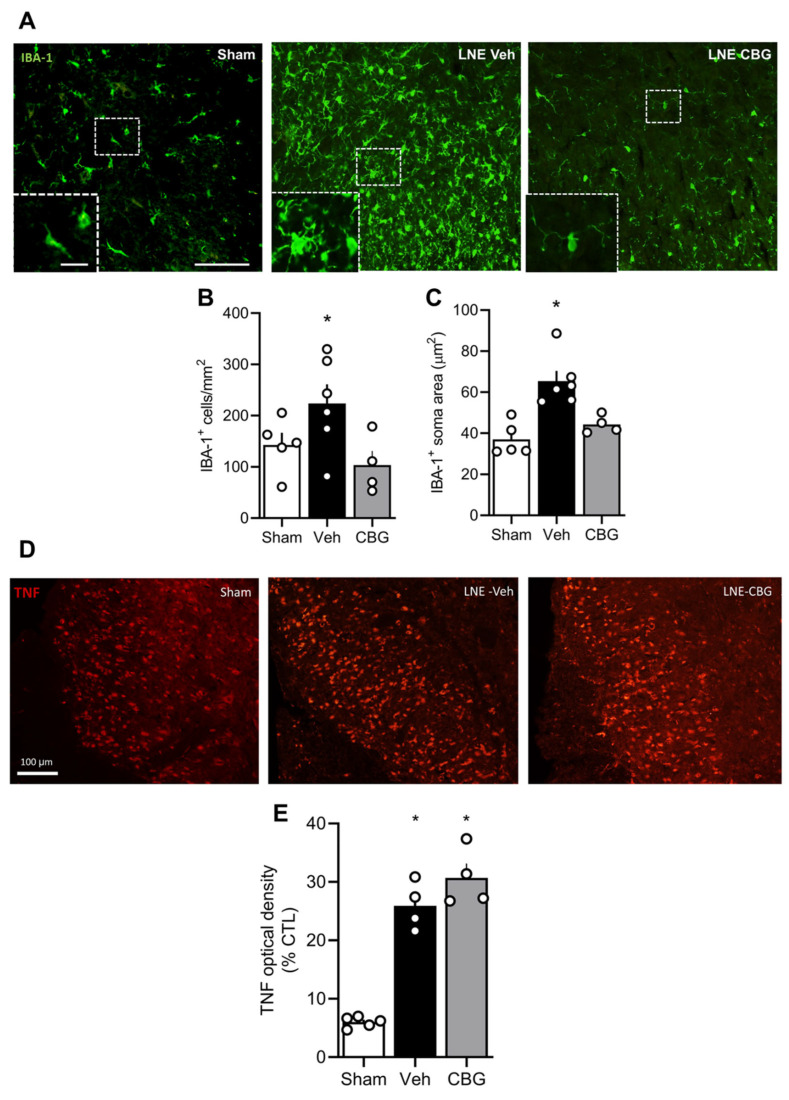
Representative image of IBA-1 stain (**A**) and TNF-alpha in (**D**) laminae I/II of spinal cord dorsal horn. Cannabigerol reduces microglia density (**B**) and soma area ((**C**); *n* = 5, 6 and 4, respectively) without affecting TNF stain ((**E**); *n* = 5, 4 and 4, respectively). Graphs are expressed as mean ± SEM. * *p* < 0.05 compared to the Sham group.

**Figure 3 pharmaceuticals-18-01508-f003:**
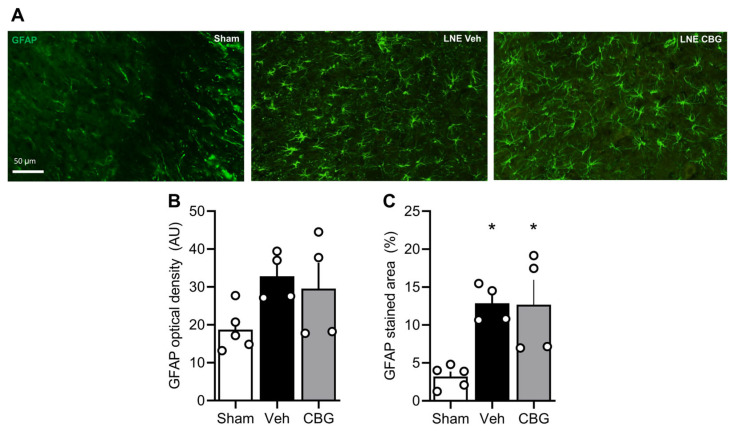
Representative image of GFAP stain (**A**) in laminae I/II of the spinal cord dorsal horn. Cannabigerol did not reverse the astrogliosis observed after SNL ((**B**,**C**); *n* 5, 4, and 4, respectively). Graphs are expressed as mean ± SEM. * *p* < 0.05 compared to the Sham group.

**Figure 4 pharmaceuticals-18-01508-f004:**
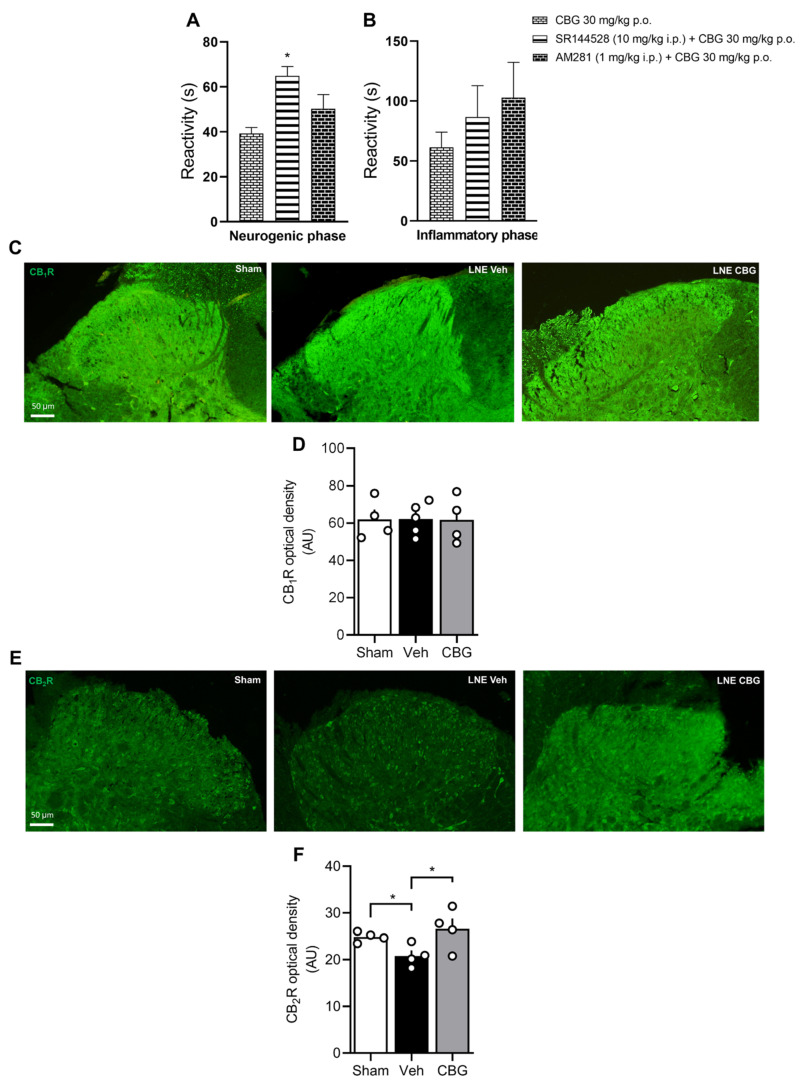
The SR144528, but not AM281, blocked the antinociceptive effect of CBG in the neurogenic phase, without affecting the inflammatory phase (**A**,**B**). Representative image of CB1R and CB2R (**C**,**E**) immunofluorescence in laminae I/II of the spinal cord dorsal horn. Cannabigerol had no effect on CB1R receptor density ((**D**); *n* = 4/group) and reversed the reduction in CB2 density observed after SNL ((**F**); *n* = 4/group). Graphs are expressed as mean ± SEM. * *p* < 0.05.

## Data Availability

The original contributions presented in this study are included in the article/Appendix A. Further inquiries can be directed to the corresponding author(s).

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
