# Peer review of "Cannabigerol Modulates Cannabinoid Receptor Type 2 Expression in the Spinal Dorsal Horn and Attenuates Neuropathic Pain Models"

_pharmaceuticals, 2025, doi:10.3390/ph18101508_

Round 1
Reviewer 1 Report
Comments and Suggestions for Authors
This manuscript, titled "Cannabigerol Modulates Nociception and Upregulates Cannabinoid Receptor Type 2 Expression in Neuropathic Pain Models," presents a timely and relevant investigation into the therapeutic potential of cannabigerol (CBG) for pain management. The study is well-structured, employing established models of acute and chronic neuropathic pain to assess the antinociceptive effects of orally administered CBG. The authors explore several potential molecular mechanisms, providing evidence for the involvement of the cannabinoid receptor type 2 (CB2R) pathway. The findings contribute valuable data to the growing body of literature on non-psychotropic cannabinoids as alternatives for pain relief. While the work is promising, several points require clarification and revision to strengthen the manuscript's conclusions.
- The novelty over existing studies with CBG is not fully clear, as prior work has already reported analgesic efficacy at similar or higher doses. The incremental advance should be explicitly clarified.
- The precise chemical composition of the extract used is not fully reported in this manuscript.
- Vehicle formulation is insufficiently described. The actual percentage of DMSO administered orally is not mentioned, and oral DMSO can influence nociceptive responses.
- Sedation or motor impairment as confounding factors are not ruled out, since locomotor assessments at the tested dose and time points are missing.
- Details on treatment randomization and blinding are insufficient. It is unclear whether allocation concealment and blinded scoring were employed.
- The CB2 antagonist blocked effects only in phase 1 of the formalin test, yet conclusions are generalized across pain modalities. This should be tempered unless further data are provided.
- In the neuropathic model, CB2 involvement is inferred mainly from immunofluorescence rather than pharmacology. Antagonist studies in this model would strengthen causal inference.
- TNF and GFAP data are interpreted as showing no effect, but without positive controls or orthogonal validation it is difficult to assess sensitivity of these assays.
- The title asserts “upregulates CB2 expression.” Given that this conclusion is based on optical density in one region and one time point, consider softening to “modulates CB2 immunoreactivity” unless corroborated by orthogonal approaches (Western/qPCR, ligand binding) and functional antagonism in SN.
- There appears to be an error in the formula provided for calculating the percentage of the maximum possible effect (%MPE) in the hot plate test. The formula presented is difficult to interpret and seems incorrect. Please review and provide the standard, correct equation.
Author Response
We thank Reviewer 1 for the careful and constructive comments, which have greatly helped us to improve the clarity and robustness of our manuscript. Below, we provide a detailed point-by-point response and indicate where changes were made.
Q1: The novelty over existing studies with CBG is not fully clear, as prior work has already reported analgesic efficacy at similar or higher doses. The incremental advance should be explicitly clarified.
A: We appreciate the reviewer’s observation. Indeed, previous studies have reported the analgesic effects of CBG, including our own earlier work showing efficacy at 50 mg/kg in a model of prenatal hypoxia–ischemia (doi:10.3390/scipharm92030053), as well as studies in chemotherapy-induced neuropathy (doi:10.3390/ph16101442.; doi:10.1002/ejp.2016). The incremental novelty of the present work lies in three aspects: (i) we demonstrate that a lower oral dose (30 mg/kg) of a CBG-rich extract is sufficient to produce robust antinociceptive effects, thereby improving translational relevance compared with earlier reports employing higher doses (≥50 mg/kg); (ii) we provide new mechanistic evidence that CBG restores CB2 receptor density reduced by spinal nerve ligation (SNL), with selective blockade by SR144528 in the neurogenic phase of the formalin test, indicating CB2-dependent activity in neuropathic pain; and (iii) beyond behavioral outcomes, we show that CBG attenuates microgliosis but not astrogliosis in the spinal dorsal horn, adding cellular insights not previously described. Collectively, these findings extend prior literature by refining the dose–response relationship, identifying CB2 signaling as a mechanistic contributor, and delineating glial modulation in neuropathic pain.
Changes in manuscript: Conclusion section (page 12, lines 411–421)
Q2: The precise chemical composition of the extract used is not fully reported in this manuscript.
A: We agree and have added the chemical composition of the extract to the Methods section; the relevant text is highlighted in lines 97-101. According to the report, the extract was highly enriched in cannabigerol (CBG, 59.0% m/m; 95%), while Δ9-THC was present only at 0.5% m/m (0.9%). Cannabidiolic acid (CBDA), Cannabidiol (CBD), Tetrahydrocannabinolic acid (THCA) and Cannabinol (CBN). were below the detection limit (<0.2%).
Q3: Vehicle formulation is insufficiently described. The actual percentage of DMSO administered orally is not mentioned, and oral DMSO can influence nociceptive responses.
A: Details of the vehicle formulation have been added to the Methods section and are now highlighted in lines 103 and 109–110. Vehicle control groups were included in all nociception tests (both acute and chronic pain models), and DMSO alone (100% v/v, ≥99.9% purity) did not produce any antinociceptive effects under the experimental conditions. As shown in the corresponding graphs, the data experimentally confirm that DMSO had no influence on nociceptive thresholds or pain-related behaviors in any of the models tested. Therefore, based on both our direct experimental previous results (PMID: 39251148 and DOI: 10.3390/scipharm92030053) and the established literature, we confidently conclude that DMSO did not exert any antinociceptive effect under the conditions tested, and that the observed effects in treated groups are solely attributable to the compound under investigation.
Q4: Sedation or motor impairment as confounding factors are not ruled out, since locomotor assessments at the tested dose and time points are missing.
A: In a previous study, a higher dose of the same extract was administered, and no signs of sedation or motor impairment were observed in the animals. Therefore, confounding effects related to sedation or motor impairment are unlikely at the dose used in the present study. ( https://doi.org/10.3390/scipharm92030053)
Changes in manuscript: Discussion section (page 10, lines 291–295)
Q5: Details on treatment randomization and blinding are insufficient. It is unclear whether allocation concealment and blinded scoring were employed.
A: We would like to confirm that the allocation of animals to the experimental groups (placebo, CBG, or reference drugs) was performed randomly to ensure there was no bias in group composition. Animals were randomly assigned to treatment groups using a random number generator before the start of experiments. Drug preparation and administration were performed by one experimenter, while behavioral scoring was conducted by a different experimenter who was blinded to treatment allocation.
Changes in manuscript: Methods section (page 3, lines 113-118).
Q6: The CB2 antagonist blocked effects only in phase 1 of the formalin test, yet conclusions are generalized across pain modalities. This should be tempered unless further data are provided.
A: We agree that our pharmacological data demonstrate CB2 receptor involvement primarily in the neurogenic phase of the formalin test, while the inflammatory phase was not fully antagonized. In light of this, we have revised our conclusions to state that CB2 receptors contribute to, rather than fully mediate, CBG’s antinociceptive effects. Importantly, our immunohistochemistry data support CB2 involvement, since CBG restored CB2 receptor density reduced by SNL, and this was paralleled by recovery of both thermal and mechanical thresholds. Together, these findings indicate that CB2 signaling plays a relevant but context-dependent role in CBG’s effects across pain modalities.
Changes in manuscript: Conclusions section (page 12, lines 411- 421).
Q7: In the neuropathic model, CB2 involvement is inferred mainly from immunofluorescence rather than pharmacology. Antagonist studies in this model would strengthen causal inference.
A: Although we infer the mechanism by the results observed in the immunofluorescence (IF) assay, we opted by not performing a pharmacological study in order to 3R recommended by the our and global Animal Committee. For instance, our IF showed a reduction in CB2R in the SNL that was reversed by CBG which may result in a higher dose of SR144528 in order to completely block CGB effect. Moreover, in order to perform a classical pharmacological assay, we firstly would need to perform a dose response curve using only the CB2R antagonist in our animal model for excluding a possible dose where SR144528 might present an intrinsic effect ( as recommended by IUPHAR that ε=0 for true antagonist). After the dose response-curve we would need to perform a new SNL assay using both CBG and SR144528 . Ultimately, this would result in the use of at least 60 new animals (10.1038/s41684-024-01476-2).
Q8: TNF and GFAP data are interpreted as showing no effect, but without positive controls or orthogonal validation it is difficult to assess sensitivity of these assays.
A: We appreciate the reviewer’s comment regarding the sensitivity of the TNFα and GFAP assays. In our study, the SNL + vehicle group serves as a positive control, as it reliably shows the expected increase in TNFα and astrogliosis compared to sham animals. This demonstrates that our immunofluorescence approach is sensitive enough to detect inflammatory changes induced by nerve injury. Importantly, immunofluorescence provides regional specificity, allowing us to evaluate TNFα and GFAP levels precisely in the dorsal horn of the spinal cord. This is a distinct advantage over whole-tissue methods such as Western blot, which assess protein levels in total tissue extracts and cannot resolve localized changes. We also note that including additional positive controls (e.g., pharmacological inducers of TNFα or astrogliosis) would increase the number of animals required without providing substantial additional insight, which would conflict with the principles of rational animal use and the 3Rs (Replacement, Reduction, Refinement). Therefore, we believe that the current experimental design is sufficient to conclude that CBG selectively modulates microglial activation but does not significantly affect TNFα upregulation or astrogliosis in the SNL model. Besides, we previously demonstrated that SNL augmented the TNF immunoreactivity in the spinal cord (10.2147/JPR.S295265) and the same animal model increased the reactivity of astrocytes in the dorsal horn, in a similar fashion observed here (10.2147/JPR.S295265). In this direction, we opted by not using another since we and others have previously consistently demonstrated the effects observed in our results, and by using another technique other animals would be necessary and it might be unethical.
Q9: The title asserts “upregulates CB2 expression.” Given that this conclusion is based on optical density in one region and one time point, consider softening to “modulates CB2 immunoreactivity” unless corroborated by orthogonal approaches (Western/qPCR, ligand binding) and functional antagonism in SN.
A: Thank you for the insight we reformulated our title to “Cannabigerol Modulates Nociception Cannabinoid Receptor Type 2 Expression in Neuropathic Pain Models”
Q9: There appears to be an error in the formula provided for calculating the percentage of the maximum possible effect (%MPE) in the hot plate test. The formula presented is difficult to interpret and seems incorrect. Please review and provide the standard, correct equation.
A: The formula for calculating %MPE in the hot plate test has been corrected to follow the standard equation used in the literature and is highlighted in line 144 and 145
Reviewer 2 Report
Comments and Suggestions for Authors
The article "Cannabigerol Modulates Nociception and Upregulates Cannabinoid Receptor Type 2 Expression in Neuropathic Pain Models" presents very interesting results in different types of pain. The introduction is adequate and up-to-date on the topic. The materials and methods are consistent with the objectives, and the figures are clear and very explanatory. However, I have some comments:
1. Why were the compounds administered 50 minutes before formalin, and when the antagonists were evaluated, they were administered 30 minutes before formalin?
2. In what volume were the compounds administered?
3. Why didn't they evaluate a positive control in neuropathic pain, such as gabapentin?
4. Could the effects of cannabigerol have a pharmacokinetic explanation, according to the literature?
Author Response
We thank Reviewer 2 for the careful and constructive comments, which have greatly helped us to improve the clarity and robustness of our manuscript. Below, we provide a detailed point-by-point response and indicate where changes were made.
Q1. Why were the compounds administered 50 minutes before formalin, and when the antagonists were evaluated, they were administered 30 minutes before formalin?
A: We appreciate the reviewer’s insightful question. The CBG extract was administered 50 minutes prior to formalin injection because this interval corresponds to the onset of its antinociceptive effect, as confirmed in the hot plate test, and is consistent with preliminary pharmacokinetic data. In contrast, the CB₂ antagonist SR144528 and the CB₁ antagonist AM281 were administered 30 minutes before formalin, a timing supported by previous studies showing effective modulation of nociceptive responses in the formalin test (PMID: 10822060; PMID: 16276190). These intervals were selected based on experimental evidence and established protocols, ensuring compound efficacy while minimizing the number of animals used, in accordance with the 3Rs.
Q2. In what volume were the compounds administered?
A: In our study, CB₁ (AM281) and CB₂ (SR144528) antagonists were administered intraperitoneally at a volume of 60 μL per mouse.
Q3. Why didn't they evaluate a positive control in neuropathic pain, such as gabapentin?
A: Gabapentin is widely used as a positive control in spinal nerve ligation (SNL) models of neuropathic pain, consistently reducing mechanical and thermal allodynia (PMID: 28450909; PMID: 23969615; PMID: 23857727). However, including an additional gabapentin-treated group in the present study would merely replicate effects already well established in the literature, without yielding proportional scientific gain. In line with the reduction principle of the 3Rs, we therefore opted not to expose additional animals to unnecessary procedures.
Q4. Could the effects of cannabigerol have a pharmacokinetic explanation, according to the literature?
A: The effects of cannabigerol (CBG) may be partially explained by its pharmacokinetics. CBG is rapidly absorbed, reaching peak plasma levels within 10–20 minutes and is metabolized by CYP2J2 (PMID: 36397993). In rats, CBG pharmacokinetics can already be characterized 30 minutes after administration (doi: 10.1007/s00213-011-2415-0), supporting the administration interval used in our study.
Reviewer 3 Report
Comments and Suggestions for Authors
The authors conducted an interesting preclinical study investigating cannabigerol (CBG) in acute and chronic neuropathic pain models. However, several limitations should be addressed before publishing this work:
1. The authors have included excessive detail and references on the epidemiology of pain, which is redundant. Instead, they should select up to three high-quality meta-analyses to support their points.
2. The sample size is small, and no sample power calculation is provided. The authors need to demonstrate that their findings are statistically valid.
3. The study is limited to male animals. Given that pain can be sex-dependent, the authors should discuss this limitation and include appropriate references regarding sex-based differences in cannabinoid responses.
4. No information is provided on the pharmacokinetics, metabolism, or potential toxicity of CBG. Data on safety and tolerability, such as weight changes, sedation, and motor side effects, are missing. Furthermore, the study tests only one dose of CBG without comparison to other cannabinoids. The authors should either present relevant results or review the literature to find appropriate references and incorporate them into the discussion.
5. The experimental design is limited, leading to overestimating claims regarding the contribution of CB2 receptor signalling to the observed effects. Additionally, they did not conduct further exploration of this signalling pathway.
6. The figures are of modest quality, with inconsistent labelling (e.g., SNL vs. LNE) and varying group sizes.
7. The roles of tumour necrosis factor (TNF) and astrogliosis are underexplored and not adequately discussed. Possible explanations for why TNF and astrogliosis were not affected may include factors such as the mechanism involved, the timing of tissue collection, regional specificity, or technical sensitivity. Additionally, the authors should clarify why CB2, but not CB1, mediates antinociception. All their claims contradict previous studies and require thorough explanations. The authors should also find relevant references and include them in the discussion.
Overall, addressing these limitations will enhance the clarity and impact of the study and may be considered for publication.
Author Response
We thank Reviewer 3 for the careful and constructive comments, which have greatly helped us to improve the clarity and robustness of our manuscript. Below, we provide a detailed point-by-point response and indicate where changes were made.
Q1. The authors have included excessive detail and references on the epidemiology of pain, which is redundant. Instead, they should select up to three high-quality meta-analyses to support their points.
A: We thank the reviewer for this helpful suggestion. In accordance with the recommendation, we have revised the Introduction and Discussion to reduce redundant details on the epidemiology and economic burden of chronic pain. Instead of multiple national estimates, we now cite up to three high-quality meta-analyses to support our points (Fayaz et al., 2016; Jackson al., 2016; Steingrímsdóttir et al., 2017). The revised text can be found in the Introduction (lines 41–49) and in the Discussion (lines 288–290) of the manuscript, and has been highlighted accordingly.
Q2. The sample size is small, and no sample power calculation is provided. The authors need to demonstrate that their findings are statistically valid.
A: We acknowledge the reviewer’s concern regarding sample size. Although the number of animals per group was relatively small in some experiments, all data were analyzed using appropriate statistical methods, and differences were considered significant when p < 0.05. Importantly, group sizes were consistent with those commonly employed in similar preclinical pain studies, ensuring statistical validity within the ethical principles of animal research. In line with the 3Rs (Replacement, Reduction, and Refinement), and as recommended by our Institutional and National Ethical Committees, we limited the number of animals to the minimum necessary to achieve scientifically meaningful results while reducing unnecessary animal use.
Q3. The study is limited to male animals. Given that pain can be sex-dependent, the authors should discuss this limitation and include appropriate references regarding sex-based differences in cannabinoid responses.
A: We agree that sex-dependent differences in pain responses are important to consider. Although our current study focused on male animals, our group has previously demonstrated that CBG can elicit distinct effects in males and females. In a prior study (https://www.mdpi.com/2218-0532/92/3/53) conducted by our group, CBG reduced TNFα expression in males and Nav1.7 expression in females, highlighting sex-specific mechanisms in nociception. This will be included in the Discussion section and noted as a limitation of the current study.
Changes in manuscript: Discussion and Limitations of the Study section (page 10 and 11, lines 382-387 and 403-406).
Q4. No information is provided on the pharmacokinetics, metabolism, or potential toxicity of CBG. Data on safety and tolerability, such as weight changes, sedation, and motor side effects, are missing. Furthermore, the study tests only one dose of CBG without comparison to other cannabinoids. The authors should either present relevant results or review the literature to find appropriate references and incorporate them into the discussion.
A: We thank the reviewer for this valuable comment. Data on CBG pharmacokinetics and metabolism have now been added to the Discussion (doi: 10.1007/s00213-011-2415-0; doi: 10.3390/pharmaceutics17020236). In our previous study, we evaluated 50 mg/kg of the same extract and found no motor alterations, supporting its tolerability. Here we tested a single effective dose (30 mg/kg), and this limitation has been acknowledged in the Discussion.
Changes in manuscript: Discussion section (page 10, lines 297-300)
Q5. The experimental design is limited, leading to overestimating claims regarding the contribution of CB2 receptor signalling to the observed effects. Additionally, they did not conduct further exploration of this signalling pathway.
A: Even though we infer the mechanism by the results observed in the immunofluorescence (IF) assay, we opted by not performing a pharmacological study in order to 3R recommended by the our and global Animal Committee. For instance, our IF showed a reduction in CB2R in the SNL that was reversed by CBG which may result in a higher dose of SR144528 in order to completely block CGB effect. Moreover, in order to perform a classical pharmacological assay, we firstly would need to perform a dose response curve using only the CB2R antagonist in our animal model for excluding a possible dose where SR144528 might present an intrinsic effect ( as recommended by IUPHAR that ε=0 for true antagonist). After the dose response-curve we would need to perform a new SNL assay using both CBG and SR144528 . Ultimately, this would result in the use of at least 60 new animals. 10.1038/s41684-024-01476-2
Q6. The figures are of modest quality, with inconsistent labelling (e.g., SNL vs. LNE) and varying group sizes.
A: Thank you for the notice. We changed the LNE for SNL.
Q7. The roles of tumour necrosis factor (TNF) and astrogliosis are underexplored and not adequately discussed. Possible explanations for why TNF and astrogliosis were not affected may include factors such as the mechanism involved, the timing of tissue collection, regional specificity, or technical sensitivity. Additionally, the authors should clarify why CB2, but not CB1, mediates antinociception. All their claims contradict previous studies and require thorough explanations. The authors should also find relevant references and include them in the discussion.
A: Thank you for the insight. The discussion now presents some other evidence from our and other groups, highlighting that astrogliosis might not be the sole player in neuropathic pain animal models (please refer to line 374 in the discussion section). Besides, we explore the relevance of CB2 by addressing the inflammatory events that usually underlie stereotypic inflammation (increased COX2 activity - line 354). And also, we believe that CB1 is not a major player in the present study since both CB1R and BDNF are unaffected by our experimental protocol, at least at the time point analyzed -where the neuropathy seems to be well consolidated (line 332).
Overall, addressing these limitations will enhance the clarity and impact of the study and may be considered for publication.
A: Thank you for your valuable feedback. Your comments will be carefully considered to improve the clarity and impact of the study.
Reviewer 4 Report
Comments and Suggestions for Authors
A well written manuscript investigating the effects of CB-enriched (?) cannabis extracts on acute and chronic pain. The results are explained, but hinge on the composition of the extract used.
This is the only critique, but unfortunately a major one. The authors do not describe the composition of their extract; cit. 20 does not specifiy the composition either besides the minor information that the strain produces less than 0,2% d9-THC. In this article, no further reference is given, but no cannabinoid analysis either. For the manuscript, an analysis of cannabinoids has to be provided.
I slightly disagree on the interpretation, that CB1R density has no effect on CBG effects – at least Fig. 4 does not contain an experiment with vehicle and may better be explained with some CB1R effects; in principle it may be compatible with many alternative effects of CB1R and CB2R.
In order to be acceptable, the authors have to provide an analysis of their specific CB extract, with quantification of all relevant cannabinoids, as well as a control for all experiments performed.
Some specific remarks:
Lines 38 – 45, and lines 278ff: Both prevalences and cost estimates for chronic pain don't match accross countries and continents – it is hard to believe 722 billion $ cost in the US, whereas in Europe with a larger population the total sum is supposed to be 12 billion $; also, 80% lost productivity in Europe and 30% in the US don't match either.
Line 79: is citation 17 correct – it doesn't mention cannabinoids nor pain.
Line 133: is there a rationale for excluding mice with a latency period of >20 sec? Possibly, they are genetically different with respect to nociception or neuropathic pain.
Methods
include a paragraph describing the composition of the specific extract used for the experiments, including the amount of THC, THC acid, CBD, CBD acid and CBN.
Figure 4: From the data, it is unclear whether CB1R antagonism has an effect or not; Fig. 4A and b indicate an effect ~ ½ of the CB2R blockade. Phrasing this as a yes/no effect does not fit the graphs.
Line 279: Sentence unclear.
Round 2
Reviewer 1 Report
Comments and Suggestions for Authors
The authors have addressed the queries raised. The manuscript may be accepted.
Reviewer 2 Report
Comments and Suggestions for Authors
The authors responded to the comments